# Dietary total antioxidant capacity of Moroccan Type 2 Diabetes Mellitus patients

**Najia El Frakchi** [1,2]*, **Khaoula El Kinany**[2], **Marwa El Baldi**[2], **Younes Saoud**[1], **Karima El Rhazi**[2]

1 Laboratory of Applied Biology and Pathology (UAE/U24FS), Faculty of Sciences, Abdelmalek Essaadi University, Tetouan, Morocco, 2 Laboratory of Epidemiology and Research in Health Sciences, Faculty of Medicine, Pharmacy and Dental Medicine, Sidi Mohammed Ben Abdellah University, Fez, Morocco

* E-mail: najia.frakchi@gmail.com

**Data Availability Statement:** All relevant data are within the manuscript.

**Funding:** The author(s) received no specific funding for this work.

## Abstract

### Aims

A new approach to a healthy diet is the assessment of dietary Total Antioxidant Capacity (TAC). The aim of this study was to assess the dietary TAC among Moroccan Type 2 Diabetes Mellitus (T2DM) patients and identify the main food sources contributing to the total antioxidant capacity intake.

### Methods

A total of 254 patients with T2DM was included in the study. The usual dietary intakes were assessed by means of a validated food frequency questionnaire. The dietary TAC was estimated using published databases of the antioxidant content of foods measured by the FRAP (ferric ion reducing antioxidant potential) method.

### Results

The mean (SD) dietary TAC of the studied type 2 diabetes patients was 10.86 (3.42) mmol/day. Correlation analyses showed a positive association between dietary TAC and the consumption of healthy food groups, such as fruits and vegetables. Tea and coffee beverages (38.6%), vegetables (21.9%), cereals and pulses (18.8%), fruits and fruit juices (12.4%) were major food sources of dietary antioxidant intake. The relatively short list of twenty food items that contributed most to dietary TAC presented an important explanation of roughly 94%. These included tea, coffee, broad beans, artichoke, pepper, beetroot, sweet potatoes, pomegranate, mandarin, figs, strawberry, orange juice, olives, cashew nuts, almonds, sunflower seeds, dchicha and white beans.

### Conclusions

This study supplies baseline dietary TAC data for Moroccan T2DM patients that may help to elucidate which aspects of the eating habits and behaviours require improvement and provide the opportunity to develop dietary guidelines as part of the nutritional diabetes management.

**Competing interests:** The authors have declared that no competing interests exist.

## Introduction

Type 2 diabetes mellitus is one of the most frequent chronic diseases worldwide and it is becoming increasingly common non-communicable disease (NCDs) in the Middle East and North Africa (MENA) region. The risk of developing cardiovascular diseases (CVDs) increases linearly with persistent hyperglycemia, favoring the emergence of coronary heart disease, atherosclerosis and other vascular complications [1, 2].

An ample evidence has shown that oxidative damage may contribute to the initiation and progression of diabetes-associated complications [3]. Chronic hyperglycemia triggers multiple metabolic pathways that lead to increased oxidative stress owing to the overproduction of free radicals [4]. Clinical studies have demonstrated accordingly higher concentrations of pro-oxidants and biomarkers of oxidative damage in T2DM patients. It has also been shown that the serum total antioxidant status decreases in T2DM [5, 6]. The low levels of antioxidant enzymes, together with the poor dietary antioxidant intake in comparison to healthy individuals can impair the antioxidant capacity in plasma of this clinical population [7].

Consumption of antioxidant-rich foods in daily diet might be protective against oxidative stress, as substantiated by their effect on increasing antioxidant capacity in the plasma [8]. Since the assessment of single antioxidant intake may not reflect the antioxidant potential of the overall diet and not consider the synergy between dietary antioxidants, a recent nutritional approach of dietary total antioxidant capacity was developed as the cumulative measure of the whole antioxidants present in foods, while considering a synergism that may exist between them [9]. It also can be considered as a predictor of diet quality and plasma antioxidant capacity [10, 11].

The link between improved diet quality and reduced risk of chronic diseases can be partly explained by dietary TAC intake [12]. In this sense, numerous studies have addressed the protective effects of high dietary TAC on metabolic disorders [13, 14]. In some studies, higher dietary TAC was associated with lower risk of hypertension [15], dyslipidemia [16] and reduced risk of incident chronic kidney disease in subjects with hyperglycemia [17]. Furthermore, diets high in TAC showed a negative association with the risk of heart failure, stroke and myocardial infarction in CVDs [18–20]. Findings from a prospective cohort studies revealed that high dietary TAC was associated with a lower risk of mortality from all causes, cancer, and CVDs [21, 22].

Morocco is known as a southern Mediterranean Sea country, and the subregion of Tetouan, located in the extreme north of Morocco, represents an interesting example of a fortified city on the Mediterranean coast. Its strategic position across the Gibraltar Gulf has made it an important crossroads between two civilizations (Spanish and Arab) and two continents (Europe and North Africa). Culturally, the typical eating habits are based on cereals, mainly soft wheat. This has increasingly substituted the traditional cereals of durum wheat and barley, particularly in the preparation of bread. Fruits and vegetables figure greatly in the diet and includes mainly citrus fruits (oranges, clementines, mandarins), apples, pears, grapes, tomatoes and tuberous vegetables. Potatoes also take an important part in the diet. The dietary pattern is also characterized by a large intake of pulses such as lentils, chickpeas, dried peas and broad beans and a greater use of olive oil. Tea still a widely consumed beverage [23]. Nonetheless, the nutritional transition towards westernised diet is in progress [24, 25].

Understanding dietary intake of antioxidants has an important interest, particularly in clinical populations with underlying oxidative stress and inflammatory injuries. The aim of this study is therefore to assess the daily dietary antioxidant capacity of T2DM patients and identify the top food contributors commonly eaten in Morocco.

## Methods

### Schematic overview of field questionnaire study

The study procedure advocates data collection in three stages (STEP) (Fig 1). The first Step consists of a face-to-face interview to collect demographic data and behavioral risk factors via a questionnaire. The second Step concerns the anthropometric parameter measurements using standard methods. The final Step includes dietary data assessment using a validated S-FFQ during a face-to-face interview by trained dietitian. This approach allows assessing dietary TAC, exploring the association between dietary TAC and the characteristics of diabetic study participants as well as identifying the major common food contributors to the antioxidant potential of the diet. This investigation may help to underline which aspects of the eating behaviors of this clinical population require improvement in order to enhance their dietary TAC and then ensuring a quality diet.

### Study participants

This cross-sectional study was carried out in Tetouan, a city in the north of Morocco, from February 2021 to July 2022 among type 2 diabetes outpatients using convenience-sampling

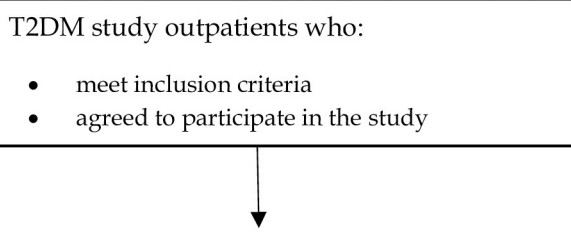

T2DM study outpatients who:

- meet inclusion criteria
- agreed to participate in the study

**Data collection**

| **Step 1** | Collecting data on basic characteristics:<br>• demographic data<br>• behavioral risk factors<br>• health status<br>Using a face-to- face interview method and a questionnaire. |
|---|---|
| **Step 2** | Anthropometric measurements:<br>• height · weight · waist circumference<br>• systolic and diastolic blood pressure<br>• Body composition analysis in terms of fat, water and muscle mass<br>Using standard methods. |
| **Step 3** | Dietary data assessment using a validated S-FFQ during a face-to-face interview by trained dietitian. |

**Fig 1. Study procedure flow diagram.**

method. The patients were enrolled from primary public health care centers and Clinical Department of Endocrinology, Diabetology at Provincial Hospital of Tetouan.

The inclusion criteria were adult patients with pathologically confirmed type 2 diabetes for at least one year and having completed data in dietary intake and confounding variables. Patients were excluded if they were pregnant, had type 1 diabetes, gestational diabetes, related disorders interfering with cognition or compliance, imperfect demographic or anthropometric information and who had missing data on dietary record. In total, a sample of 254 T2DM patients aged 34–76 years was examined in the present study.

We should highlight that the majority of diabetes patients were women. This finding is supported by previous studies which reported that T2DM is a predominant disease in women [26, 27]. Additionally, the National Family Health Survey assumes that a female gender presents a prominent risk group [28].

Participation was voluntary, and informed written consents were required from all participants. The study protocol was approved by the ethical committee of the University Hospital Center of Fez (ref. n˚ 06/20).

## Demographic data

A face-to-face interview was carried out to collect information on sociodemographic and lifestyle features such as age, sex, marital status, educational level, occupation and physical activity using an epidemiological questionnaire. It was based on the STEPS instrument developed by the World Health Organization (WHO) according to the STEPwise approach for chronic disease survey and which was also used in the National Survey on Common Risk Factors for NCDs [29, 30].

Educational level was classified as "illiterates", "≤ 6 years of schooling" (primary, informal or koranic education), and "> 6 years of schooling "(secondary, university level). Two categories of the marital status were included: married and singles, while "singles" were widowed, unmarried and divorced. Occupation activity was recorded into two classes: "worker" and "retired or unemployed or housekeeper".

Physical activity level was assessed with the version translated to Arabic of Global Physical Activity Questionnaire (GPAQ) [31]. Data on physical activity were analysed according to the GPAQ Analysis Guide [32] and the metabolic equivalent task (MET) value was computed. Physical activity level was categorised as low (< 600 MET-minutes per week), moderate (600–3000 MET-minutes per week) and high level (≥ 3000 MET-minutes per week) [33].

## Clinical measurements

The clinical measurements were performed by healthcare personnel according to the standardized procedures [34]. Body weight was measured with a calibrated electronic scale in minimal clothing and without shoes. Standing height without shoes was also obtained using a mural stadiometer. Body mass index (BMI) was calculated as weight in kilograms divided by the square of the height in meters [35]. Systolic and diastolic blood pressure was measured using an automatic Omron tensiometer for three times and the mean of the last two blood pressure measurements was considered in the analyses. Arterial hypertension in the studied diabetes patients was defined as taking antihypertensive medication or systolic blood pressure ≥ 140 mmHg and/or diastolic blood pressure ≥ 90 mmHg [36].

## Dietary intake assessment

The usual dietary intake of the studied diabetes patients was assessed using a validated semi-quantitative food frequency questionnaire (FFQ) with 255 food items, which was done face-to-

face by trained dietitian. As previously described [37], this FFQ has shown a good relative validity compared to the dietary average obtained from three 24-hour recalls and there were no sex differences regarding the reliability of reported dietary intakes. The questionnaire was validated for use in Moroccan population and was designed to obtain data on usual food intake during the previous year. Food consumption for each item was questioned based on a daily, weekly, or monthly frequency versus portion size. The reported frequency was then converted to the amount of each several consumed food items in grams per day. Energy and nutrient intakes of consumed foods were calculated using the food composition tables from Tunisia [38] and Morocco [39].

## Assessment of dietary TAC

Dietary TAC was estimated using a previously published database with the total amount of antioxidant capacity expressed as mmol per 100 g of over 3100 types of foods and beverages from different countries [40, 41], obtained according to ferric-reducing antioxidant power (FRAP) method [42]. The dietary antioxidant capacity was estimated by multiplying the daily intake of foods and beverages by their related antioxidant content. Thus, the dietary TAC is a sum of all TAC values from each food item consumed, in mmol per day. For foods for which TAC values were not available on the database, the value of the closest comparable foodstuff was used as a proxy: for bakkoula (Moroccan salad recipe based on leaves similar to spinach) the TAC value for spinach has been used; for dchicha (traditional Moroccan soup based on barley semolina) and saykouk (Moroccan dish made from barley couscous and buttermilk), the estimated TAC for the barley was used. For cooked foods with missing FRAP values, the value of the fresh food was considered. Finally, the dietary TAC estimate was classified into three groups (tertiles). The first and third tertiles had the lowest and highest dietary TAC scores, respectively.

## Statistical methods

The SPSS Statistics for Windows, version 25.0 was used to perform the Statistical analyses. The normality of variables was tested using the Kolmogorov-Smirnov test. Continuous variables were presented as means ± standard deviation and categorical variables were presented as numbers and percentages. Variables were categorized by tertiles of dietary TAC. The chi-square test was used to compare categorical data, and the analysis of variance (ANOVA) or the Kruskal-Wallis test was used to compare continuous data.

Correlation analyses between dietary TAC and dietary variables with normal and non-normal distribution were done respectively by the Pearson and Spearman test.

The contribution of different food groups to the total dietary antioxidant intake was computed as the ratio of the antioxidant capacity provided by the corresponding food group to the dietary TAC estimate from all foods.

To assess the explanatory ability of various foods and beverage items on FRAP levels; we used stepwise multiple regression analysis. A $p$ for entry into the regression model of $< 0.05$ and a $p$ for removal of $> 0.10$ were used. We first ran separate models within each food group (vegetables, cereals, fruits, legumes, dairy product, oils, nuts and beverage) to identify which food items that best explained the dietary antioxidant variation within a food group. In the final model, we included all the individual food items obtained from the preceding analysis. We report both the standardized and the unstandardized regression coefficients. To select the top food predictors, the list of the food items was figured in descending order according to the beta values.

All reported *p*-values are based on two-tailed hypotheses and *p*-values < 0.05 were regarded significant.

## Results

### Estimated daily dietary TAC

The mean (SD) dietary antioxidant capacity was 10.86 (3.42) mmol/day, and the mean energy intake was 2561.9 (643.7) kcal/day.

The general characteristics of participants by FRAP indices tertiles are displayed in Table 1. The majority of diabetes patients were women and from urban areas. The mean (SD) of age was 54.52 (7.21) years old and the mean (SD) diabetes duration was 8.2 (6.4) years. The prevalence of overweight and obesity was, respectively, 39.8% and 47.2%, while arterial hypertension prevailed in 70% of the studied diabetes patients.

As can be seen, participants with a higher dietary antioxidant capacity had significantly a lower age (*p* = 0.036), higher diabetes duration (*p* = 0.018), lower systolic blood pressure level (*p* = 0.008), and were more likely to be physically active than those with lower values of TAC (*p* = 0.01). No significant differences were found for the remaining variables.

### Dietary intake according to dietary antioxidant capacity level

Nutrient and food intakes across tertiles of the energy-adjusted dietary TAC of patients with T2DM are presented in Table 2. In comparison with the participants assigned to the lowest tertile, those in the highest tertile of the dietary TAC had significantly higher intakes of carbohydrate (*p* = 0.012), total protein (*p* = 0.009), polyunsaturated fatty acid (*p* = 0.041), dietary fiber (*p* = 0.002), calcium (*p* < 0.001), magnesium (*p* < 0.001), iron (*p* = 0.022), folic acid (*p* < 0.001), thiamin (*p* < 0.001), riboflavin (*p* < 0.001), niacin (*p* = 0.003), vitamin B6 (*p* = 0.003), and some nutrients with antioxidant potential, including vitamin A (*p* = 0.028), vitamin C (*p* = 0.002), β-carotene (*p* = 0.029), and zinc intakes (*p < 0.001*).

In addition, participants in the highest tertile of dietary TAC had higher consumption of vegetables (*p* < 0.001), fruits (*p* = 0.041), tea (*p* < 0.001) and coffee (*p* = 0.012), than those included in the lowest tertile. No other significant differences were found in food and nutrient intakes across tertiles of energy-adjusted dietary TAC.

As observed in correlation analysis in Table 2, dietary TAC showed a statistically significant positive correlation with the estimate intakes of vitamin C (r = 0.212; *p* = 0.001), thiamin (r = 0.141; *p* = 0.025), riboflavin (r = 0.204; *p* = 0.001), folic acid (r = 0.192; *p* = 0.002), magnesium (r = 0.284; *p* <0.001), calcium (r = 0.204; *p* = 0.001), iron (r = 0.129; *p* = 0.041), zinc (r = 0.157; *p* = 0.012) and the consumption of vegetables (r = 0.335; *p* < 0.001), fruits (r = 0.159; *p* = 0.011), tea and coffee beverages (r = 0.465, r = 0.217; respectively; *p* < 0.001). However, there was a negative correlation between the dietary TAC and the consumption of refined grain cereals (r = − 0.13, *p* = 0.038).

### Major food sources contributing to the dietary antioxidants

The main food groups that contributed to the dietary antioxidant capacity intake were tea and coffee beverages (38.6%), vegetables (21.9%), cereals and pulses (18.8%), fruits and fruit juices (12.4%). The other food categories, such as potatoes, nuts and seeds, fish, dairy products, meats and olive oil, contributed altogether only to around 7% of antioxidants intake (Fig 2).

To identify the contribution of each food item consumed to the dietary TAC variation of the diabetes patient's diet, all the several individual food categories were forced into the stepwise multiple regressions with dietary TAC as further described. The regression coefficients

**Table 1. General characteristics of T2DM patients according to tertiles of dietary TAC.**

| Variables | All | Dietary Total Antioxidant Capacity | | | p- value |
|---|---|---|---|---|---|
| | | T1 (lowest) | T2 | T3 (highest) | |
| | | (n = 85) | (n = 84) | (n = 85) | |
| **Gender**, | | | | | 0.609 |
| Woman, n (%) | 219(86.2) | 72(84.7) | 75(89.3) | 72(84.7) | |
| **Age** (years), Mean ± SD | 54.52 ± 7.21 | 55.73 ± 6.17 | 54.32 ± 8.09 | 53.49 ± 7.14 | 0.036 |
| **Marital status**, n (%) | | | | | 0.969 |
| Married | 207(81.5) | 70(82.4) | 68(81) | 69(81.2) | |
| Singles | 47(18.5) | 15(17.6) | 16(19) | 16(18.8) | |
| **Occupation**, n (%) | | | | | 0.208 |
| Worker | 53(20.9) | 14(16.5) | 16(19) | 23(27.1) | |
| Retired or unemployed or house keeper | 201(79.1) | 71(83.5) | 68(81) | 62(72.9) | |
| **Residency** | | | | | 0.311 |
| Urban, n (%) | 237(93.3) | 77(90.6) | 81(96.4) | 79(92.9) | |
| **Monthly household income**, n (%) | | | | | 0.094 |
| < 3000 MAD | 210(82.7) | 77(91.7) | 69(83.1) | 64(80) | |
| > 3000 MAD | 37(14.6) | 7(8.3) | 14(16.9) | 16(20) | |
| **Educational level**, n (%) | | | | | 0.912 |
| Illiterate | 117(46.1) | 41(48.2) | 39(46.4) | 37(43.5) | |
| ≤ 6 years | 98(38.6) | 30(35.3) | 34(40.5) | 34(40) | |
| > 6 years | 39(15.4) | 14(16.5) | 11(13.1) | 14(16.5) | |
| **Diabetes duration** (years), | 8.2 ± 6.4 | 7.2 ± 7.1 | 8.2 ± 5.7 | 9.1 ± 6.1 | 0.018 |
| Mean ± SD | | | | | |
| **Medication use**, n (%) | | | | | 0.122 |
| Oral Antidiabetic Drugs | 176(69.3) | 60(70.6) | 65(77.4) | 51(60) | |
| Insulin | 51(20.1) | 14(16.5) | 15(17.9) | 22(25.9) | |
| Oral Antidiabetic Drugs + Insulin | 19(7.5) | 8(9.4) | 4(4.8) | 7(8.2) | |
| **Hypertension**, n (%) | 177(70) | 66(77.6) | 54(65.1) | 57(67.1) | 0.16 |
| **Dyslipidemia**, n (%) | 90(35.4) | 28(32.9) | 31(36.9) | 31(36.5) | 0.839 |
| **Physical activity**, n (%) | | | | | 0.175 |
| Low | 77(30.3) | 33(38.8) | 24(28.6) | 20(23.5) | |
| Moderate | 66(26.0) | 23(27.1) | 20(23.8) | 23(27.1) | |
| High | 111(43.7) | 29(34.1) | 40(47.6) | 42(49.4) | |
| **Physical activity**. METs-h per week | 50.49 ± 46.72 | 39.7 ± 36.2 | 50.4 ± 39.5 | 61.2 ± 58.9 | 0.010 |
| **Body mass index** (kg/m$^2$), n (%) | | | | | 0.157 |
| Normal | 33(13.0) | 7(8.2) | 9(10.7) | 17(20) | |
| Overweight (25–30) | 101(39.8) | 33(38.8) | 34(40.5) | 34(40) | |
| Obesity (≥30) | 120(47.2) | 45(52.9) | 41(48.8) | 34(40) | |
| **Blood pressure** (mmHg), Mean ± SD | | | | | |
| Systolic Blood pressure | 140.75 ± 20.01 | 145.87 ± 19.10 | 139.82 ± 20.69 | 136.57 ± 19.31 | 0.008 |
| Diastolic Blood pressure | 80.53 ± 11.23 | 81.98 ± 12.72 | 80.36 ± 10.20 | 79.22 ± 10.55 | 0.36 |

Continuous variables are presented as mean ± SD, while categorical variables are presented as number of the participants (percentages).

$p$ -value is for ANOVA or Kruskal-Wallis test for continuous variables and Chi-square test for a categorical variable, $p < 0.05$ was considered as statistically significant.

MAD Moroccan Dirham, MET, metabolic equivalent task

**Table 2. Dietary characteristics according to the tertiles of dietary TAC of T2DM patients.**

| | Dietary total antioxidant capacity [†] | | | | |
|---|---|---|---|---|---|
| | Coefficient correlation (r) | T1 (< 9.84) | T2 (9.84–12.24) | T3 (> 12.24) | p [‡] |
| **Nutrients intake** | | | | | |
| Carbohydrate (g/day) | 0.004 | 312.43 ± 100.0 | 346.6 ± 88.9 | 312.45 ± 82.26 | 0.012 |
| Protein (g/day) | 0.039 | 84.2 ± 25.5 | 95.5 ± 24.8 | 87.6 ± 22.4 | 0.009 |
| Total fat (g/day) | − 0.030 | 109.2 ± 39.7 | 117.7 ± 39.9 | 115.9 ± 37.7 | 0.334 |
| SFA [e] (g/day) | 0.008 | 19.3 ± 8.5 | 21.1 ± 7.3 | 19.7 ± 7.1 | 0.090 |
| MUFA [c] (g/day) | 0.008 | 52.4 ± 19.3 | 55.6 ± 21.3 | 56.1 ± 20.2 | 0.398 |
| PUFA [d] (g/day) | 0.011 | 17.7 ± 6.3 | 19.8 ± 6.5 | 18.0 ± 5.7 | 0.041 |
| Fiber (g/day) | 0.109 | 46.7 ± 19.2 | 56.4 ± 20.8 | 50.4 ± 18.5 | 0.002 |
| Folic acid (mcg/day) | 0.192 [a] | 880.1 ± 346.0 | 1199.5 ± 506.6 | 1164.1 ± 439.4 | < 0.001 |
| Vitamin A (mcg/day) | 0.105 | 479.3 ± 214.0 | 591.6 ± 299.8 | 555.0 ± 258.5 | 0.028 |
| Vitamin E (mg/day) | 0.070 | 20.2 ± 7.6 | 22.8 ± 8.4 | 22.2 ± 7.4 | 0.050 |
| Vitamin C (mg/day) | 0.212 [b] | 86.9 ± 42.7 | 100.6 ± 44.0 | 112.1 ± 55.5 | 0.002 |
| Vitamin D (mcg/day) | 0.062 | 6.3 ± 3.2 | 6.8 ± 3.2 | 7.0 ± 3.4 | 0.473 |
| Vitamin B6 (mg/day) | 0.117 | 57.0 ± 88.4 | 83.7 ± 116.1 | 61.0 ± 68.8 | 0.003 |
| Vitamin B12 (mcg/day) | 0.079 | 40.1 ± 36.1 | 41.3 ± 39.5 | 46.2 ± 36.7 | 0.292 |
| Riboflavin (mg/day) | 0.204 [b] | 2.2 ± 0.6 | 2.7 ± 0.7 | 2.6 ± 0.7 | < 0.001 |
| Niacin (mg/day) | 0.096 | 23.6 ± 6.2 | 27.0 ± 6.1 | 25.9 ± 6.7 | 0.003 |
| Thiamin (mg/day) | 0.141[a] | 2.4 ± 0.6 | 2.9 ± 0.8 | 2.8 ± 0.8 | < 0.001 |
| β-carotene (mcg/day) | 0.119 | 3253.8 ± 2424.3 | 4152.7 ± 2449.0 | 4030.4 ± 2924.5 | 0.029 |
| Magnesium (mg / day) | 0.284 [b] | 541.1 ± 173.3 | 705.3 ± 242.3 | 708.2 ± 234.8 | < 0.001 |
| Calcium (mg/day) | 0.204 [b] | 623.2 ± 203.8 | 767.7 ± 274.5 | 774.6 ± 264.1 | < 0.001 |
| Iron (mg/day) | 0.129 [b] | 49.9 ± 41.6 | 62.2 ± 47.3 | 72.7 ± 65.9 | 0.022 |
| Zinc (mg/day) | 0.157 [b] | 25.2 ± 16.6 | 34.6 ± 21.6 | 30.4 ± 14.5 | < 0.001 |
| Selenium (mcg/day) | 0.007 | 129.2 ± 35.6 | 139.6 ± 32.3 | 132.5 ± 35.6 | 0.126 |
| **Food intakes. g/day** | | | | | |
| Vegetables | 0.335 [b] | 225.0 ± 97.9 | 340.6 ± 154.4 | 362.5 ± 164.5 | < 0.001 |
| Potatoes | − 0.008 | 71.1 ± 57.7 | 75.8 ± 43.8 | 68.3 ± 53.0 | 0.074 |
| Legumes | − 0.060 | 126.7 ± 94.5 | 153.1 ± 120.8 | 118.7 ± 89.2 | 0.146 |
| Cereals | − 0.085 | 388.0 ± 115.8 | 407.5 ± 97.9 | 368.7 ± 107.9 | 0.064 |
| Whole grain cereals | 0.091 | 180.4± 169.5 | 196.1 ± 171.2 | 207.2 ± 156.2 | 0.406 |
| Refined grain cereals | − 0.130 [b] | 209.2 ± 164.4 | 218.3 ± 165.0 | 168.1 ± 141.2 | 0.133 |
| Fruits | 0.159 [b] | 187.6 ± 129.2 | 196.0 ± 109.0 | 231.9 ± 132.0 | 0.041 |
| Fruit juices | 0.093 | 31.2 ± 44.8 | 40.8 ± 51.6 | 43.2 ± 70.2 | 0.332 |
| Dairy and dairy products | 0.015 | 86.0 ± 72.6 | 105.4 ± 94.7 | 105.4 ± 115.9 | 0.537 |
| Red meat | − 0.100 | 17.01 ± 18.66 | 17.06 ± 19.18 | 13.46 ± 16.29 | 0.37 |
| Poultry | − 0.049 | 30.3 ± 23.5 | 31.1 ± 21.1 | 28.6 ± 19.8 | 0.77 |
| Eggs | − 0.064 | 16.8 ± 12.7 | 21.0 ± 25.6 | 15.7 ± 13.3 | 0.56 |
| Fish | 0.098 | 45.1 ± 25.9 | 48.3 ± 25.0 | 53.6 ± 27.7 | 0.16 |
| Tea | 0.465 [b] | 97.1 ± 84.4 | 163.5 ± 115.4 | 274.2 ± 186.2 | < 0.001 |
| Coffee | 0.217 [b] | 63.1 ± 57.6 | 88.9 ± 78.9 | 127.3 ± 131.1 | 0.012 |
| Nuts and seeds | 0.117 | 1.6 ± 3.2 | 4.0 ± 7.1 | 4.6 ± 11.0 | 0.168 |

*(Continued)*

**Table 2.** (Continued)

| | Dietary total antioxidant capacity [†] | | | | |
|---|---|---|---|---|---|
| | Coefficient correlation | T1 | T2 | T3 | *p* [‡] |
| | (r) | (< 9.84) | (9.84–12.24) | (> 12.24) | |
| Fats and Oils | − 0.028 | 49.1 ± 22.1 | 49.1 ± 24.4 | 50.2 ± 22.7 | *0.829* |

[†]Adjusted for Energy intake by residual method. T, tertile. TAC (mmol/day) Mean (SD) minimum—maximum: T1 (8.15 [1.19], 5.59–9.83); T2 (10.94 [0.72], 9.84–12.24); T3 (14.68 [2.06], 12.25–20.96).

[‡] *p*—values is for ANOVA for continuous variables with normal distribution or Kruskal-Wallis test for continuous variables without normal distribution

[a] *p* -value <0.05 according to Pearson's Correlation test (r),

[b] *p* -value <0.05 according to Spearman's Correlation test (r)

Data are expressed as means ± standard deviations, *p* <0.05 was considered as statistically significant.

[c] MUFA, monounsaturated fatty acid;

[d] PUFA, polyunsaturated fatty acid;

[e] SFA, saturated fatty acid

from the final model was presented in Table 3. The top twenty food items predicting the dietary antioxidant intake were listed according to the decreasing impact. These foods were tea and coffee beverages, broad beans, artichoke, almonds, orange juice, dchicha (traditional Moroccan soup based on barley semolina), pomegranate, olives, mandarin, pepper, sweet potatoes, beetroot, sunflower seeds, white beans, strawberry, figs and cashew nuts. These 20 food items explained 94% of the total antioxidant intake variation. When we reran the model with excluding tea and coffee contribution, the rest of the selected food items remain as important explanatory of roughly 54%.

## Discussion

In the present study, we assessed the dietary TAC in T2DM patients. The food sources that contributed most to the total antioxidant capacity intake were also identified. Up to our

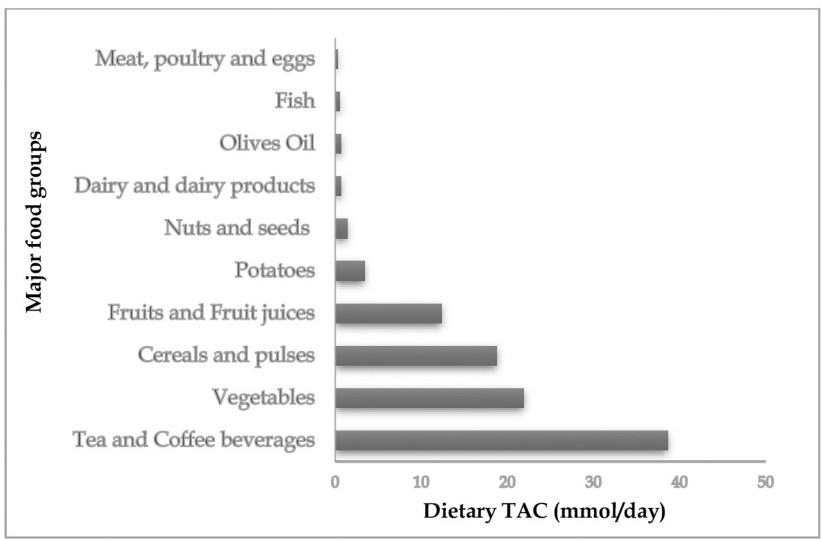

**Fig 2. Major food sources of dietary TAC of Moroccan T2DM patients.**

**Table 3. Regression coefficients of top 20 food items contributing to antioxidant intake in T2DM patients.**

| Rank | Food items* (g/day) | Unstandardized coefficients | | Standardized coefficients | |
|------|---------------------|-----------|-----------|----------|---------|
| | | Estimate | Std. error | Estimate | p- value |
| 1 | Tea, green, prepared | 0.01 | < 0.001 | 0.63 | < 0.001 |
| 2 | coffee, liquid | 0.02 | 0.001 | 0.49 | < 0.001 |
| 3 | Broad beans, green | 0.02 | 0.001 | 0.39 | < 0.001 |
| 4 | Coffee, instant, prepared | 0.02 | 0.001 | 0.19 | < 0.001 |
| 5 | Black tea, infusion | 0.01 | 0.001 | 0.19 | < 0.001 |
| 6 | Artichoke, boiled | 0.06 | < 0.01 | 0.14 | < 0.001 |
| 7 | Almonds | 0.16 | 0.02 | 0.13 | < 0.001 |
| 8 | Orange juice | < 0.01 | 0.001 | 0.12 | < 0.001 |
| 9 | Dchicha | 0.02 | < 0.01 | 0.12 | < 0.001 |
| 10 | Pomegranate | 0.02 | < 0.01 | 0.11 | < 0.001 |
| 11 | Olives | 0.01 | < 0.01 | 0.10 | < 0.001 |
| 12 | Mandarin | 0.02 | < 0.01 | 0.08 | < 0.001 |
| 13 | Pepper | 0.02 | < 0.01 | 0.07 | < 0.001 |
| 14 | Sweet potatoes | 0.01 | < 0.01 | 0.07 | < 0.001 |
| 15 | Beetroot | 0.03 | < 0.01 | 0.06 | < 0.001 |
| 16 | Sunflower seeds | 0.06 | 0.02 | 0.06 | < 0.001 |
| 17 | White beans | < 0.01 | 0.001 | 0.06 | < 0.001 |
| 18 | Strawberry | 0.04 | 0.01 | 0.04 | 0.01 |
| 19 | Figs | 0.02 | < 0.01 | 0.04 | 0.02 |
| 20 | Cashew nuts | 0.22 | 0.10 | 0.04 | 0.03 |

* Ranked in descending order according to standardized beta coefficients.

knowledge, this is the first attempt to explore dietary antioxidant index levels among Moroccans. Nonetheless, the adherence to a Mediterranean dietary pattern was widely assessed [43, 44].

The estimated dietary TAC of Moroccan patients with type 2 diabetes using the FRAP method was 10.86 mmol/day. This finding is in accordance with TAC of 10.62 mmol/day in patients with metabolic syndrome from Spain [45], the TAC of 11.86 mmol/day in adults with prediabetes from Persian country [46] and to the TAC of 10.84 mmol/day in patients with cardiovascular disease [47]. Meanwhile, some observational studies showed that consumption of dietary TAC approximately $\geq$ 13.48 mmol/d based on the FRAP assay was associated with lower risk of cardiovascular disease [48]. Findings from a meta-analysis of prospective cohort studies have shown that a 5 mmol/day increase in dietary TAC, based on the FRAP method, was associated with a 7% lower risk of all-cause mortality [21].

As regards the distribution of descriptive features of diabetes patients by dietary TAC (Table 1), the finding that diabetic patients with low physical activity behavior and high blood pressure level consumed a diet with a lower antioxidant potential, could be a serious health concern. Physical inactivity and elevated blood pressure are common risk factors associated with CVDs in diabetes patients. Therefore, in these specific conditions, a greater average of antioxidant intake is recommended since the increased demand. In addition, it was found that diabetes patients in the highest tertile of dietary TAC had a greater intake of dietary fibre and higher consumption of plant-based foods such as fruits and vegetables than those in the lowest tertile (Table 2). The benefits of high fiber diet in T2DM has been long appreciated. A high intake of dietary fibre will benefit the metabolic health in patient with T2DM [49]. An inverse

relationship between total dietary fiber intake and the risk of developing CVDs has been also evidenced [50].

Some authors suggest that dietary TAC may be considered an appropriate measure of diet quality because it positively correlates with well-known indicators of a healthy diet [51]. Those indicators approve mutually high intake of healthy foods such as fruits and vegetables, which provide a good source of antioxidants, and then contribute to the antioxidant potential of the diet. Our study supported this association, showing a positive relationship between dietary FRAP and the consumption of vegetables and fruits as well (Table 2).

A strong correlation has been found between dietary TAC and plasma antioxidant capacity [11], suggesting that dietary antioxidant capacity might be investigated as a measurement of antioxidant intakes. The present study showed a significant positive association between dietary TAC and dietary intake of vitamin C, folic acid, iron, magnesium and zinc (Table 2), in agreement with previous findings [14, 52].

There is a wide divergence between countries regarding food contribution to the dietary TAC intake. The main contributors to the dietary antioxidant capacity in Greek population were fruits (45.5%), vegetables (38.8%), dry fruits (31.1%), cereals (28%) and nuts (26.1%) [53]. In Spanish population, there was coffee (43.7%), fruits (26.2%) followed by vegetables [54]. The Major contributors to dietary FRAP in French women were fruits (23%), vegetables (19%), alcoholic beverages (15%) and hot beverages such as tea and hot chocolate (12%) [55]. The major foods in Italian population were coffee, alcoholic beverages, fruit and fruit juices [56]. In the Polish population with cardiovascular disease, there were tea and coffee beverages, vegetables, fruits and cereal products [57]. The food groups that contributed most to dietary FRAP in the Rotterdam population were coffee, fruit, vegetables and tea [58]. The main contributors to the dietary TAC in the Turkish population were beverages, vegetables, legumes/nuts and fruits [59]. In our study, the highest contributors to dietary TAC estimate were tea and coffee beverages, vegetables, fruits, cereals and pulses (Fig 2). As expected, coffee and tea drinks stood out as the major contributor to dietary antioxidant both due to their high antioxidant level and their consumption by a large part of the study population. The group of fruits and vegetables was the second that most contributed to the determination of dietary antioxidant capacity in the studied diabetes patients. Despite the fact that these foods are nutrient-rich and antioxidant- abundant, it may also be partially explained by the appropriate rate of fruit and vegetable consumption. The mean daily consumption of vegetables (without potatoes) and fruits estimated was 309.6 g and 214.7g, respectively, values well as recommended by the WHO [60].

There will often be a significant difference in which food explains the total intake, and which explain the between person variation. Identifying the dietary factors contributing to the dietary antioxidant variation is more important for research in the epidemiology of diseases. Whereas, assessing the foods that contribute to total antioxidant intake can be useful for clinical nutrition, which aims to focus on good sources with high antioxidant capacities and commonly eaten in order to improve the antioxidant capacity in the patient's diet. Therefore, this study also addressed capturing the antioxidant intake variation (Table 3). Coffee and tea drinks were found to have the greatest impact on the dietary TAC. Fruits and vegetables also contributed significantly to the variation. Among vegetables, the most important item was broad beans and among fruits, it was the pomegranate. However, when we removed the tea and coffee beverages from the list of top 20 food items explaining most the antioxidant intake variation, the remaining fruit and vegetable foodstuffs contributed 48.6%. In other words, compared to the contribution of coffee and tea, fruits and vegetables explain in the same extent of around 49% the between person variation in antioxidant intake.

The current study has some limitations. The dietary TAC determination was based on an international database, in which the values may vary in relation to the food produced in Morocco. Therefore, we selected database that reported total antioxidant content of large food items consumed worldwide and which contained most of the foods that are commonly consumed by the Moroccan population. The strength of this study is that it is the first attempt to assess the total antioxidant potential of antioxidants consumed in the whole diet by the concept of dietary TAC index among Moroccan patients with T2DM. Furthermore, the data collection was performed by qualified nutritionist, and the questionnaire used, even though is prone to measurement error, it was detailed and validated for use in the Moroccan adult population, then capturing most of the antioxidant-rich foods.

## Conclusion

In summary, the estimate of the dietary TAC of Moroccan patients with T2DM was found closely comparable to that of the population of Mediterranean countries and those with similar metabolic disorders. Nonetheless, dietary TAC intake in this clinical population needs to be further improved. Tea, coffee, vegetables, fruits and cereals were the major food sources of antioxidant capacity of the diet. Whereas, despite the tea and coffee beverages, broad beans, artichoke, pomegranate, orange juice, almonds, dchicha, olives and mandarin were among the major foods contributing in dietary TAC. Therefore, consumption of these foods may be a good strategy to increase dietary TAC. This study supplies baseline dietary TAC data among Moroccan T2DM patients, which will be useful in developing dietary recommendations as part of the nutritional management of diabetes. Further investigation of the health contribution of dietary antioxidant capacity in diabetes outcome in Morocco is warranted.

## Acknowledgments

The authors appreciate the cooperation of the diabetes patients in this work.

## Author Contributions

**Conceptualization:** Najia El Frakchi, Karima El Rhazi.

**Data curation:** Najia El Frakchi, Khaoula El Kinany, Marwa El Baldi, Karima El Rhazi.

**Formal analysis:** Najia El Frakchi, Khaoula El Kinany, Marwa El Baldi, Karima El Rhazi.

**Methodology:** Najia El Frakchi, Khaoula El Kinany, Karima El Rhazi.

**Project administration:** Younes Saoud.

**Supervision:** Younes Saoud, Karima El Rhazi.

**Validation:** Younes Saoud, Karima El Rhazi.

**Writing – original draft:** Najia El Frakchi.

**Writing – review & editing:** Najia El Frakchi.

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
