## [Decision Letter · Decision Letter 0]

10 Dec 2023

PONE-D-23-22322Dietary Total Antioxidant Capacity of subjects with Type 2 Diabetes Mellitus in Northern MoroccoPLOS ONE

Dear Dr. EL FRAKCHI,

Thank you for submitting your manuscript to PLOS ONE. After careful consideration, we feel that it has merit but does not fully meet PLOS ONE’s publication criteria as it currently stands. Therefore, we invite you to submit a revised version of the manuscript that addresses the points raised during the review process.

**ACADEMIC EDITOR:**Please kindly check the comments below.

We look forward to receiving your revised manuscript.

Kind regards,

Charles Odilichukwu R. Okpala

Academic Editor

PLOS ONE

Journal Requirements:

3. We noticed that this work is related to the following submission of yours currently under review: PONE-D-23-22223 (Association of dietary total antioxidant capacity to general and abdominal obesity in patients with Type 2 Diabetes Mellitus in Northern Morocco.)

Upon submission, authors must confirm that the manuscript, or any related manuscript, is not currently under consideration or accepted elsewhere. If related work has been submitted to PLOS ONE or elsewhere, authors must include a copy with the submitted article (as 'Other' file). Please also discuss the related submission in your cover letter. Reviewers will be asked to comment on the overlap between related submissions (http://journals.plos.org/plosone/s/submission-guidelines#loc-related-manuscripts).

Additional Editor Comments:

Please, reviewers have found your work very promising, and have provided useful comments.

Kindly attend to them diligently, and provide sufficient details

Look forward to your revised manuscript.

Reviewers' comments:

Reviewer's Responses to Questions

**Comments to the Author**

1. Is the manuscript technically sound, and do the data support the conclusions?

Reviewer #1: Yes

Reviewer #2: Partly

2. Has the statistical analysis been performed appropriately and rigorously? 

Reviewer #1: Yes

Reviewer #2: Yes

3. Have the authors made all data underlying the findings in their manuscript fully available?

Reviewer #1: Yes

Reviewer #2: Yes

4. Is the manuscript presented in an intelligible fashion and written in standard English?

Reviewer #1: Yes

Reviewer #2: No

5. Review Comments to the Author

Reviewer #1: The manuscrciprt tilted "Dietary Total Antioxidant Capacity of subjects with Type 2 Diabetes Mellitus in Northern Morocco" is an interesting investigation.

In general, the work was methodical, well written and easy to follow.

However, the authors should consider the following suggestions:

1. Edit to improve the quality of Figure 1 by showing the line axes and also remove the gridlines.

2. Include a section on Conclusion in the Manuscript

3. It would be helpful to have a control (perhaps non diabetic patient) in this investigation to compare.

Reviewer #2: The present manuscript is a good attempt to document the dietary antioxidant intake through validated food frequency questionnaire using the documented international data of FRAP content of foods. The authors should clearly state the primary outcome and secondary outcome of the study. The sampling methodology and the research design needs to be clearly spelt out. The inclusion and exclusion criteria are not stated clearly. The statistical treatment to the antioxidant value data with nutrients and food groups is adequate but it does not states clearly discusses the connect between Dietary Antioxidant intake and Type 2 diabetes.

6. PLOS authors have the option to publish the peer review history of their article (what does this mean?). If published, this will include your full peer review and any attached files.

Reviewer #1: No

Reviewer #2: No

---

## [Author Response · Author response to Decision Letter 0]

2 Feb 2024

Request:

In response to your request, I would like to inform you that I have submitted simultaneously to PLOS ONE journal two original research articles that have been written as an independent paper for publication.

The aforementioned manuscripts were the fruit of a survey carried out on the same clinic population of Moroccan patients with T2DM. Although the same methodology was used for data collection and assessment of dietary intake including dietary TAC assessment, the objective of each manuscript was different.

In PONE-D-23-22322 manuscript, we aimed to bring to light new information about the daily intake of antioxidants among patients with T2DM on the basis of the dietary antioxidant index since it is the first study assessing dietary TAC in Moroccan country. The contribution of common food eaten in Morocco to total antioxidant capacity intake was also examined.

In PONE-D-23-22223 manuscript, we aimed to explore the relationship between dietary TAC of these patients and obesity-related features regarding the importance of obesity in this clinic population as a common comorbidity of type 2 diabetes and one of the most modifiable risk factors for preventing other comorbid conditions, such as cardiovascular disease.

We think so that it is desirable to publish separately the above works and not to present the full data within a single manuscript since each one deals with an objective. Also, that may help to provide more easily understandable information.

We declare that the manuscript is original, has not been published before, and is not currently under consideration or accepted elsewhere.

Reviewer 1: 

The quality of Figure 1 was improved as recommended and conclusion section was included as well.

Agreed. Further case-control designed investigation may be considered thereafter.

Reviewer 2: 

I carefully rewrite the study design and population section with the intent to meet the aforementioned requirements.

I have also added a short sentence to discuss the connect between Dietary Antioxidant intake and Type 2 diabetes through presented data on dietary intake according to dietary antioxidant capacity.

---

## [Decision Letter · Decision Letter 1]

20 Feb 2024

PONE-D-23-22322R1Dietary Total Antioxidant Capacity of subjects with Type 2 Diabetes Mellitus in Northern MoroccoPLOS ONE

Dear Dr. EL FRAKCHI,

Thank you for submitting your manuscript to PLOS ONE. After careful consideration, we feel that it has merit but does not fully meet PLOS ONE’s publication criteria as it currently stands. Therefore, we invite you to submit a revised version of the manuscript that addresses the points raised during the review process.

**ACADEMIC EDITOR:**Please see comments below.

We look forward to receiving your revised manuscript.

Kind regards,

Charles Odilichukwu R. Okpala

Academic Editor

PLOS ONE

Journal Requirements:

Additional Editor Comments:

Please, authors kindly attend to the comments raised by a reviewer

You will see that reviewers agree that your work is worthy of publication

The editor also suggests the following:

a) In the introduction, please squeeze in this information, before line 91, and some of it before line 98: 1) The food-health situation of the subregion where Morocco is situated , the foods that dominate the region culturally, the demographics of urban and rural areas within the subregion, and why it is reflective of the dietary situation found in Morrocco

b) Please methods, start this section with new subsection captioned 'Schematic overview of field questionnaire study', which should comprise 4-5 sentences and supported by a flow diagram that shows us how you arranged the entire study. Sentence 1 should introduce the title of flow diagram, Sentence two should talk about the key stages, sentence 3 how it directly connects with the objective of the study, and sentence 4 and 5 show why the study is relevant, why such data analysis was employed, etc. Make sure this new subsection captures all the key stages of the methods

c) I can see you separated your results and discussion. Please, kindly make effort to state (Refer to Table ?) or (Refer to Figure ?) in all the places where tables/figure captured in results and indicated in the discussion. So, all the tables and figure must be captured. Please, make sure to do this, to guide the readers.

d) Please, in your conclusions, provide some recommendations for future work

Look forward to your revised manuscript. Thanks very much :)

Reviewers' comments:

Reviewer's Responses to Questions

**Comments to the Author**

1. If the authors have adequately addressed your comments raised in a previous round of review and you feel that this manuscript is now acceptable for publication, you may indicate that here to bypass the “Comments to the Author” section, enter your conflict of interest statement in the “Confidential to Editor” section, and submit your "Accept" recommendation.

Reviewer #1: All comments have been addressed

Reviewer #2: All comments have been addressed

2. Is the manuscript technically sound, and do the data support the conclusions?

Reviewer #1: Yes

Reviewer #2: Yes

3. Has the statistical analysis been performed appropriately and rigorously? 

Reviewer #1: Yes

Reviewer #2: Yes

4. Have the authors made all data underlying the findings in their manuscript fully available?

Reviewer #1: Yes

Reviewer #2: Yes

5. Is the manuscript presented in an intelligible fashion and written in standard English?

Reviewer #1: Yes

Reviewer #2: Yes

6. Review Comments to the Author

Reviewer #1: (No Response)

Reviewer #2: All the desired and suggested changes are made. One observation in Table 2 about the unit for quantity of food intake especially green tea, black tea, instant coffee in gm per day is not acceptable and other foods consumed preferably should also be defined as number of serving per day which equate to the Total Antioxidant Content in the diet.

7. PLOS authors have the option to publish the peer review history of their article (what does this mean?). If published, this will include your full peer review and any attached files.

Reviewer #1: No

Reviewer #2: **Yes: **Jagmeet Madan

---

## [Author Response · Author response to Decision Letter 1]

19 Mar 2024

Response to an Additional Editor Comments:

First, I would like to thank you for your consideration of this manuscript.

a) We have added the additional information.

b) The new subsection 'Schematic overview of field questionnaire study' is inserted

c) Rightly, we make the corresponding reference of table and figure in the text

d) Further research is recommended as well

Response to a Request:

In response to your request, I would like to inform you that I have submitted simultaneously to PLOS ONE journal two original research articles that have been written as an independent paper for publication.

The aforementioned manuscripts were the fruit of a survey carried out on the same clinic population of Moroccan patients with T2DM. Although the same methodology was used for data collection and assessment of dietary intake including dietary TAC assessment, the objective of each manuscript was different.

In PONE-D-23-22322 manuscript, we aimed to bring to light new information about the daily intake of antioxidants among patients with T2DM on the basis of the dietary antioxidant index since it is the first study assessing dietary TAC in Moroccan country. The contribution of common food eaten in Morocco to total antioxidant capacity intake was also examined.

In PONE-D-23-22223 manuscript, we aimed to explore the relationship between dietary TAC of these patients and obesity-related features regarding the importance of obesity in this clinic population as a common comorbidity of type 2 diabetes and one of the most modifiable risk factors for preventing other comorbid conditions, such as cardiovascular disease.

We think so that it is desirable to publish separately the above works and not to present the full data within a single manuscript since each one deals with an objective. Also, that may help to provide more easily understandable information.

We declare that the manuscript is original, has not been published before, and is not currently under consideration or accepted elsewhere.

Response to a Reviewer 1: 

The quality of Figure 1 was improved as recommended and conclusion section was included as well.

Agreed. Further case-control designed investigation may be considered thereafter.

Response to a Reviewer 2: 

I carefully rewrite the study design and population section with the intent to meet the aforementioned requirements.

I have also added a short sentence to discuss the connect between Dietary Antioxidant intake and Type 2 diabetes through presented data on dietary intake according to dietary antioxidant capacity.

---

## [Decision Letter · Decision Letter 2]

24 Mar 2024

Dietary Total Antioxidant Capacity of Moroccan Type 2 Diabetes Mellitus patients

PONE-D-23-22322R2

Dear Dr. EL FRAKCHI,

We’re pleased to inform you that your manuscript has been judged scientifically suitable for publication and will be formally accepted for publication once it meets all outstanding technical requirements.

Kind regards,

Charles Odilichukwu R. Okpala

Academic Editor

PLOS ONE

Additional Editor Comments (optional):

Thank you for revising your work. It is now acceptable for publication.

Reviewers' comments:

Reviewer's Responses to Questions

**Comments to the Author**

1. If the authors have adequately addressed your comments raised in a previous round of review and you feel that this manuscript is now acceptable for publication, you may indicate that here to bypass the “Comments to the Author” section, enter your conflict of interest statement in the “Confidential to Editor” section, and submit your "Accept" recommendation.

Reviewer #2: All comments have been addressed

2. Is the manuscript technically sound, and do the data support the conclusions?

Reviewer #2: (No Response)

3. Has the statistical analysis been performed appropriately and rigorously? 

Reviewer #2: (No Response)

4. Have the authors made all data underlying the findings in their manuscript fully available?

Reviewer #2: (No Response)

5. Is the manuscript presented in an intelligible fashion and written in standard English?

Reviewer #2: (No Response)

6. Review Comments to the Author

Reviewer #2: (No Response)

7. PLOS authors have the option to publish the peer review history of their article (what does this mean?). If published, this will include your full peer review and any attached files.

Reviewer #2: **Yes: **Jagmeet Madan

---

## [Editor Report · Acceptance letter]

2 Apr 2024

PONE-D-23-22322R2 

PLOS ONE

Dear Dr. El Frakchi, 

I'm pleased to inform you that your manuscript has been deemed suitable for publication in PLOS ONE. Congratulations! Your manuscript is now being handed over to our production team.

Kind regards, 

on behalf of

Dr. Charles Odilichukwu R. Okpala 

Academic Editor

PLOS ONE